# The Hepatic Pre-Metastatic Niche

**DOI:** 10.3390/cancers14153731

**Published:** 2022-07-31

**Authors:** Benjamin Ormseth, Amblessed Onuma, Hongji Zhang, Allan Tsung

**Affiliations:** 1Department of Surgery, Division of Surgical Oncology, The Ohio State University Wexner Medical Center, Columbus, OH 43210, USA; benjamin.ormseth@osumc.edu (B.O.); amblessed.onuma@osumc.edu (A.O.); hongji.zhang@osumc.edu (H.Z.); 2Department of Surgery, Division of Surgical Oncology, The University of Virginia University Hospital, Charlottesville, VA 22903, USA

**Keywords:** pre-metastatic niche, liver, exosomes, bone-marrow-derived cells, metastasis, immunosuppression, ECM remodeling

## Abstract

**Simple Summary:**

The pre-metastatic niche is a recently established concept that could lead to targeted therapies that prevent metastasis before ever occurring. Considering that 90% of cancer mortality results from metastasis, the PMN is thus a salient opportunity for intervention. The purpose of the current review is to cover what is known specifically about the hepatic pre-metastatic niche, a topic that has garnered increasing research focus within the last decade. We discuss the methods of communication between primary tumors and the liver, the involved cell populations, the key changes within liver tissue, and perspectives on the future of the field.

**Abstract:**

Primary tumors can communicate with the liver to establish a microenvironment that favors metastatic colonization prior to dissemination, forming what is termed the “pre-metastatic niche” (PMN). Through diverse signaling mechanisms, distant malignancies can both influence hepatic cells directly as well as recruit immune cells into the PMN. The result is a set of changes within the hepatic tissue that increase susceptibility of tumor cell invasion and outgrowth upon dissemination. Thus, the PMN offers a novel step in the traditional metastatic cascade that could offer opportunities for clinical intervention. The involved signaling molecules also offer promise as biomarkers. Ultimately, while the existence of the hepatic PMN is well-established, continued research effort and use of innovative models are required to reach a functional knowledge of PMN mechanisms that can be further targeted.

## 1. Introduction

The liver is a major hub of metastasis for primary tumors from around the body. Gastrointestinal malignancies such as colon, pancreatic, and gastric carcinomas have a high tendency for hepatic spread and for breast carcinomas, sarcomas, and melanoma as well [1]. Considering that metastasis is responsible for 90% of all cancer-related deaths, decreasing spread to the liver could thus offer a significant clinical benefit for patients with varying cancer types [2]. At present, however, few such therapeutics exist. The complexity of the metastatic cascade poses a significant challenge to this end, and there is an ongoing need for further elucidation to discover new targets for intervention.

Recent evidence has begun to shed light on a new aspect of cancer biology termed the “pre-metastatic niche” (PMN). The concept was first posed by Stephen Paget’s 1889 “seed and soil” theory, which speculated that circulating tumor cells of distinct cancer origin (the seeds) had a higher predilection to spread to organs with compatible characteristics (the soil) [3]. Work within the last two decades has expanded on this theory and shown that primary tumors can prime sites of future metastasis via signaling pathways that precede tumor cell dissemination. In 2005, a landmark study by Kaplan et al. was the first to give credence to the PMN by showing that VEGFR+ hematopoietic progenitor cells accumulated in pre-metastatic lung tissue and activated resident fibroblasts [4]. 

Although research on the PMN has primarily focused on the lung since this time, a significant body of work now exists regarding the pre-metastatic liver as well. While clinical recognition of the PMN is not yet feasible, it is likely that gastrointestinal malignancies that predominate liver metastases are the same cancers which favor hepatic PMN formation. In line with this notion, the bulk of primary literature that has begun to elucidate the hepatic PMN has used colorectal carcinoma and pancreatic ductal adenocarcinoma models. The current dynamic can be thought of as a game among three players: the primary tumor of interest, the liver, and the systemic immune population of bone-marrow-derived cells (BMDCs). Signaling initiated by the primary tumor causes varying interactions among these three entities that ultimately result in principle changes that define the PMN: ECM remodeling, inflammation, immune suppression, and increased vascular permeability. 

In this review, we will discuss the signaling mechanisms, cellular components, and the characteristic changes involved in the hepatic PMN. Thereafter, we will present perspectives on the PMN targets and biomarkers that may one day offer clinical utility as well as emerging models that might lead to further discovery on the mechanisms underlying the PMN. A schematic of the salient components to the hepatic PMN can be seen in Figure 1.

## 2. Primary-Tumor-Derived Molecular Components

To influence the distant hepatic PMN, primary tumors must be able to communicate remotely through the blood stream. This is accomplished through the production and secretion of molecular components that either influence the PMN directly or instigate the recruitment of BMDCs to the niche, where they then have further effects. Thus, primary tumor signaling serves as the first step in PMN formation. The following sections will describe the known signaling mechanisms utilized and their downstream targets, and the molecular components have been summarized in Table 1.

### 2.1. Secreted Molecules

One secreted molecule that has been studied extensively is the tissue inhibitor of metalloproteinases (TIMP)-1. Seubert et al. demonstrated that TIMP-1 released by lymphoma cells could recruit CXCR4+ neutrophils to the liver by upregulating SDF-1 in the PMN [5]. Subsequent study showed that SDF-1 upregulation was mediated by TIMP-1 activation of hepatic stellate cells (HSCs) [6]. Moreover, Kopitz et al. showed that artificially increasing TIMP-1 in fibrosarcoma models shifts metastasis from the lung to the liver. Herein, TIMP-1 induced a pro-invasive and pro-proliferative microenvironment by increased hepatocyte growth factor (HGF) signaling and downstream expression of metastasis promoting genes [7]. Clinically, TIMP-1 serum levels have been shown to be inversely correlated with survival in PDAC patients, offering a motivation for continued research into TIMP-1 as a driver of metastasis via influence on the PMN [8].

Beyond TIMP-1, other molecular factors have been implicated in the development of the PMN. In a colorectal carcinoma (CRC) model, Wang et al. demonstrated that tumor-derived VEGFA stimulated tumor-associated macrophages (TAMs) in the surrounding stroma to produce CXCL1. In the blood stream, CXCL1 then recruited CXCR2+ myeloid-derived suppressor cells (MDSCs) from the bone marrow to the hepatic PMN, where they facilitated formation of liver metastases (mechanisms of MDSC involvement to be reviewed in a later section) [9]. Furthermore, in two additional studies, CXCR2 inhibition reduced MDSC and neutrophil accumulation in the PMN [10], and VEGFA inhibition with TSU68 mitigated neutrophil and macrophage recruitment [11]. Collectively these findings underscore the broad immune impact of VEGFA. Another secreted factor, granulocyte stimulating factor (G-CSF), was shown to shown to mobilize of Ly6G+Ly6C+ granulocytes in a breast cancer model. In this study, the mobilized immune cells produced the Bv8 protein, which has been implicated in angiogenesis in prior study. This protein then went on to stimulate tumor cell migration through the prokinectin receptor (PKR)-1, ultimately leading to enhanced metastatic ability [12].

Primary tumor cells can also independently secrete chemokines to further develop the PMN. Human CRC cell lines injected into mouse spleens were shown to secrete CCL15, which then recruited CD34+ Gr-1^−^ immature myeloid cells to the pre-metastatic liver through an interaction with CCR1 [13]. Antagonism of this receptor blocked myeloid accumulation into the liver, reduced metastases, and prolonged survival. A later study determined that CCL15 is upregulated following a loss of SMAD4 expression. From a clinical standpoint, they also found threefold more CCR1+ myeloid cells in human livers bearing CCL15+ metastases, and these patients had significantly shorter times of disease-free survival [14]. CCL2 is another tumor-derived chemokine implicated in the early metastatic cascade. Not only has it been shown to increase vascular permeability within the primary tumor microenvironment to augment dissemination [15], but systemic secretion was shown to recruit myeloid cells to the liver and facilitate further colonization [16]. However, in this model, myeloid recruitment occurred predominantly after initial tumor cell invasion into the liver; thus, CCL2 involvement at the pre-metastatic stage is not yet certain.

### 2.2. Exosomes

Exosomes are extra-cellular vesicles (EVs) that contain varying components of proteins, lipids, RNA, and/or DNA that can be horizontally transferred to other cells. These EVs are surrounded by a lipid bilayer that gives them protection and enables their travel throughout the circulatory system. Their diameter ranges from 30–150 nm, and they can be actively shed by cells to communicate with others both locally and distally. Depending on the integrins expressed on their surface, they can also deliver their cargo in a cell-specific manner [17]. 

Due to their unique composition and characteristics, exomes engage in cell communication throughout a broad range of physiologic functions and pathologies. Of particular interest is their involvement within the metastatic cascade. Ranging from promoting an inflammatory vascular niche in brain metastases to facilitating breast cancer dormancy, they have been implicated in both the development of the primary tumors along with progression of established metastases. Their function within these contexts has been reviewed extensively [18]; in this review, the focus will remain on their influence in forming the hepatic PMN.

By secreting exosomes into the bloodstream for hepatic uptake, primary tumors can manipulate the liver from afar before ever seeding it. Non-small cell lung cancer can secrete exosomes containing regulatory microRNA-122-5p that increases the expression of mesenchymal markers N-cadherin and Vimentin in native hepatocytes upon uptake while decreasing the epithelial marker E-cadherin. The ensuing migratory phenotype with less cellular adhesion proteins may ultimately make the liver more amenable to tumor cell invasion [19]. CRC exosomes can also stimulate hepatocytes to release hepatocyte growth factor (HGF) by suppressing SPINT1 expression. This then promotes stromal cell proliferation and migration, which increases chances of successful circulating tumor cell colonization [20].

Exosomes can also be utilized to modulate stromal cells directly. In a study by Li et al., gastric tumor cells could upregulate TGF-β1 transcription in Kupffer cells via cell-specific exosome signaling. Consequently, TGF-β1 expression in the hepatic niche activated the SMAD2/3 pathway in invading tumor cells, which increased their stem cell-like properties and led to a higher metastatic burden [21]. In addition, hepatic uptake of PDAC-secreted exosomes led to increased deposition of fibronectin in the microenvironment as well as increased recruitment of bone-marrow-derived macrophages to liver [22]. 

While their further roles in the characteristic changes of the PMN will be reviewed in a later section, their involvement can already be appreciated. From directly altering hepatocytes to engaging distant immune cells, exosomes offer several opportunities for therapeutic targeting. Special attention in future study should be paid to this form of pre-metastatic communication.

**Table 1 cancers-14-03731-t001:** The list of primary-tumor-derived molecular components and associated mechanisms involved in the development of the hepatic PMN.

Tumor Secreted Factor	Primary Tumor	Target	Mechanism	References
TIMP-1	PDAC	CD63+ HSCs	Stimulated HSCs to secrete SDF1, leading to recruitment of CXCR4+ neutrophils to the liver	[5,6]
TIMP-1	Lymphoma (L-CI.5s)	Liver parenchyma	Induced HGF signaling in the liver and downstream upregulation of metastasis-associated genes, including *HGF* and genes encoding HGF-activating proteases	[7]
VEGFA/CXCL1	CRC	TAMs	Stimulated TAMs in the primary tumor microenvironment to release CXCL1 in the blood that subsequently recruited CXCR2+ MDSCs to the PMN	[9]
G-CSF	Breast (4T1)	Ly6G + Ly6C + Granulocytes	Mobilized Ly6G+Ly6C+ granulocytes to the liver, where they produced Bv8, which increased tumor cell migration	[12]
CCL15	CRC	CD34 + Gr-1^−^ immature myeloid cells (iMCs)	Mobilized CCR1+ CD34+ Gr-1^−^ iMCs to the liver, where they produced MMP2 and MMP9	[13]
CCL2	CRC	CD11b/GR1^mid^ myeloid cells	Mobilized CCR2+ CD11b/GR1^mid^ myeloid cells to the liver	[16]
EV CCL2	CRC	Macrophages	Recruited macrophages to the liver and induced M2 phenotype polarization along with increased liver fibrosis	[23]
EV miR-122-5p	NSCLC	Hepatocytes	Stimulated hepatocyte upregulation of N-cadherin and Vimentin along with a downregulation of E-cadherin	[19]
EV miR-221/222	CRC	Hepatocytes	Activated liver HGF by suppressing SPINT1 expression	[20]
EV miR-151a-3p	Gastric cancer	Kupffer cells	Stimulated TGF-ß1 activation in Kupffer cells leading to SMAD2/3 pathway activation and enhanced stemness of incoming gastric cancer cells	[21]
EV MIF	PDAC	Kupffer cells	Stimulated TGF-ß1 activation in Kupffer cells leading to upregulation of fibronectin production by HSCs	[22]
EV miR-92a	Lewis lung carcinoma	HSCs	Secreted by BMDCs, with suppressed *SMAD7* leading to upregulation of TGF-ß signaling in HSCs, which increased ECM deposition	[24]
EV miR-181a-5p	CRC	HSCs	Activated HSCs through the IL6/STAT3 pathway leading to tumor-associated ECM deposition and secretion of CCL20	[25]
Integrin α_v_ß_5_+ exosomes	PDAC	Kupffer cells	Activated Src phosphorylation and pro-inflammatory *S100* gene expression	[26]
EV miR-21-5p	CRC	Kupffer cells	Bound to TLR6 on Kupffer cells and induced polarization into the proinflammatory phenotype	[27]
EV ANGPTL1	CRC	Kupffer cells	Decreased MMP9 secretion by Kupffer cells through JAK2-STAT3 inhibition	[28]
EV TGF-ß1	Breast	LSECs	Induced LSEC endothelial to mesenchymal transition and upregulation of fibronectin	[29]
EV TGF-ß1	PDAC	NK cells	Reduced NKG2D, CD107a, TNF-α, and INF-γ expression in NK cells, leading to decreased cytotoxicity against pancreatic cancer cells	[30]
EV lncRNA-ALAHM	Lung adenocarcinoma	Hepatocytes	Stimulated hepatocyte HGF parasecretion by binding with AUF1	[31]
EV CD44v6/C1QBP	PDAC	HSCs	Phosphorylated HSC insulin-like growth factor 1 (IGF-1) signaling molecules, leading to increased liver fibrosis	[32]
EV ITGBL1	CRC	HSCs	Stimulated TNFAIp3-mediated NF-κB signaling to activate HSCs, which then secreted proinflammatory IL-6 and IL-8	[33]
KC	CRC	CD11b+ Gr-1^−^ myeloid cells	Recruited CD11b+ Gr-1^−^ myeloid cells to the liver, where they exhibited immunosuppressive effects	[34]
IL-6	CRC	CD14+ HLA-DR^-/low^ MDSCs	Recruited MDSCs to the liver, which inhibited autologous T-cell proliferation	[35]
GRP78	Breast (E0771)	Dendritic cells, Kupffer cells	Inhibited dendritic cell activation in the liver, induced M2-like polarization of Kupffer cells, and enhanced TGF-ß production	[36]
EV miR-135a-5p	CRC	T-cells	Inhibited CD30-mediated T-cell activation to facilitate immune tolerance in the liver	[37]
EV miR-25-3p	CRC	LSECs	Increased vascular leakiness within the liver by targeting LSEC KLF2 and KLF4, leading to increased VEGF2A and decreased ZO-1, occluding, and Claudin-5 expression	[38]
EV mi-638	HCC	LSECs	Decreased expression of VE-cadherin and ZO-1	[39]

## 3. Bone-Marrow-Derived Cells

Bone-marrow-derived cells (BMDCs) are integral to the development of the hepatic PMN. They are mobilized to the liver via primary tumor stimulation, where they coordinate a metastasis-friendly microenvironment through a variety of mechanisms. Each subpopulation has corresponding capabilities and has been implicated in unique ways, but collectively, BMDCs comprise a third cellular component outside of the primary tumor or liver that is vital to PMN priming. 

### 3.1. Myeloid-Derived Suppressor Cells

Myeloid-derived suppressor cells (MDSCs) are one of the most influential populations of cells to the development of the PMN and have been implicated in the metastatic progression of many primary cancer types. They comprise a heterogeneous group of cells derived from myeloid progenitors that have powerful immunosuppressive properties along with roles in inflammation, angiogenesis, and ECM remodeling. The two primary sub-populations are split between monocytic MDSCs and granulocytic MDSCs. While they are part of the normal-functioning immune system, pathologic activation via primary cancers can recruit these cells to distant microenvironments and co-opt their function for cancer progression [40].

Previous research in contexts outside of the PMN have uncovered a number of different mechanisms by which MDSCs can stifle the natural immune response. Through production of nitric oxide (NO) and reactive oxygen species (ROS), they can suppress cytotoxic T-cell function by increasing antigen-specific tolerance and disrupting the T-cell receptor [41]. They also secrete arginase I, which depletes L-arginine necessary for T-cell proliferation and cytokine production [42]. In addition, they secrete indoleamine 2,3-dioxygenase (IDO) to degrade available tryptophan [43] and produce peroxynitrate to mitigate cytotoxic T-cell entry into tumors via nitrating chemokines [44]. Moreover, they can directly sequester nutrients from the ECM such as L-cysteine and upregulate PD-L1 expression in Kupffer cells [45,46]. 

Beyond their direct roles, MDSCs also influence other components of the immune system. They are able to induce NK cell anergy, thereby reducing clearance of migrating cancer cells. Additionally, they can augment the function of T-regulatory cells, which are known immunosuppressors [47,48]. Another pro-metastatic MDSC function could be the release of matrix metalloproteinases MMP2 and MMP9, which remodel the ECM to be more susceptible to cancer cell invasion. Additionally, MDSCs have been shown to increase vascular leakiness although neither of these studies assessed these effects at the pre-metastatic phase specifically [13,49]. Further detail on their mechanisms of action can be found in an extensive review on MDSCs by Kumar et al. [50].

In terms of spatiotemporal localization, MDSCs could already be found in the blood of mice with pre-malignant pancreatic neoplasms, with their concentration only increasing with further oncogenesis [51]. This suggests the development of the PMN might even co-occur with the timeline of the primary tumor itself. Ichikawa et al. also directly demonstrated MDSC accumulations within hepatic PMNs of CRC-bearing mice compared to controls [52]. Neither study examined by which specific mechanisms MDSC promoted metastatic colonization in the liver although it is likely a mixture of the mechanisms mentioned above.

Ultimately, MDSCs are critical players in the development of the hepatic PMN. While there has been much study on the function of these cells and their roles in liver metastasis, still there is a dearth of evidence that specifically investigates their mechanisms of action during the elusive pre-metastatic timeframe. Not only does this present an opportunity for further study, but the ensuing knowledge might offer therapeutic targets early in the metastatic cascade, which could halt the development of clinical liver masses before detectable colonization.

### 3.2. Macrophages

Another population of BMDCs mobilized to the liver are macrophages. While resident macrophages known as Kupffer cells exist in healthy liver stroma, during metastasis, a similar yet distinct population of leukocytes of monocytic origin travel into the hepatic niche from the circulatory system. Like MDSCs, they can generally promote metastasis via several different mechanisms including angiogenesis, immunosuppression, and ECM remodeling although whether they behave similarly in the PMN is unknown. What is known, however, is that they do accumulate in the liver prior to metastasis and promote subsequent colonization [53]. This recruitment was shown by Sanford et al. to depend on CCL2 release from primary PDAC tumors. 

One point of interest about tumor-associated macrophages (TAMs) is their inherent plasticity. They can differentiate into subpopulations with vastly different consequences on inflammation and immune regulation. Depending on their incoming cytokine signaling, they can either become an M1-type with anti-microbial, anti-tumor cytotoxic properties, or they can become an M2-type that promotes immunosuppression and tissue remodeling, which favor tumorigenesis [54]. While the M1-type activates TH1 T-cells and amplifies the anti-tumor response, M2-type macrophages secrete TGF-B and IL-10, which are known anti-inflammatory proteins that can induce Tregs and further propagate the immunosuppressive effect [55].

Takano et al. demonstrated this plasticity within the context of hepatic metastases using a CRC model. Tumor-derived EVs containing microRNA-203 caused monocyte polarization into the M2 phenotype. In vivo, this led to increased liver metastasis although the question remains whether this similarly happens in the PMN [56]. Clinical studies have shown an increased M2/M1 ratio in CRC patients with higher numbers of hepatic metastases compared to controls, suggesting the balance of their polarization is correlated with metastatic potential [57]. Moreover, even after differentiation, macrophages can reprogram towards the reverse phenotype [58]. If such plasticity could be harnessed within reference to the PMN, therapeutics could instigate an anti-tumor microenvironment within the liver more likely to clear disseminated cancer cells upon arrival. In reality, macrophages exist on a spectrum between the opposing phenotypes, and although they present an attractive opportunity, the mechanisms behind their PMN involvement are poorly understood. Before therapeutic development can be considered, further study is required to bridge these gaps.

### 3.3. Neutrophils

Neutrophils are another key element of the immune system under investigation for their involvement in liver metastasis. Traditionally, neutrophils are seen as first responders of the innate immune system that clear pathogens and inflammatory debris through induction of phagocytosis, production of ROS, and release of lytic enzymes [59]. However, their influence on inflammation implicates them in a wide variety of contexts, and oncogenesis is no exception. In primary tumor tissues, they have previously been shown to support tumor growth, invasion, and angiogenesis possibly due to release of ECM remodeling enzymes such as MMP-8, MMP-9, elastase, and cathepsin G [60]. Much like the aforementioned macrophages, neutrophils have also been shown to exhibit N1 and N2 sub-phenotypes with contrasting effects on tumorigenesis, and mouse models of breast cancer demonstrated preferential polarization of neutrophils within liver metastases towards the N2 type [61,62]. In a similar CRC model, polarized neutrophils within mets were linked to increased vascular density and branching, suggesting an important role in angiogenesis [63].

As mentioned previously, CXCR4+ neutrophils were shown to accumulate in hepatic PMN following primary tumor TIMP-1 secretion [5]. Alternatively, the CXCL1/CXCR2 signaling pathway has also been implicated. As demonstrated by Yamamoto, inhibiting CXCR2 both reduced neutrophil accumulation and metastatic development [11]. The actual mechanisms promoting metastasis were further explored by Hirai et al. Only one day after injection of CRC cells into mouse spleens, neutrophils began to accumulate in the pre-metastatic liver, where they secreted ECM-remodeling proteins MMP2 and MMP9. Moreover, their recruitment was linked to further accumulation of other immune cells such as monocytes and fibrocytes, suggesting neutrophil involvement in multiple downstream pathways [64].

As was the case with tumor-associated macrophages, the further mechanisms by which neutrophils facilitate the PMN within the liver need to be elucidated. One unique avenue to explore relates to their capacity to release neutrophil extracellular traps (NETs), which are extracellular DNA webs that have previously been shown to trap circulating lung carcinoma cells within hepatic sinusoids and increase micro-metastases [65]. Because the study involved artificial stimulation of NET formation, it is unclear whether human tumors can stimulate neutrophils the same way.

### 3.4. NK Cells

In addition to the populations mentioned above, NK (natural killer) cells are another immune cell type with probable involvement in the PMN. They are highly abundant in the liver and play salient roles in mitigating cancer progression in several malignancies including hepatocellular carcinoma [66]. This is in large part due to their interaction with dendritic cells (DCs), which can affect levels of NK activation and anti-tumor capacities [67,68]. In terms of pre-metastatic involvement, pancreatic cancer models have been shown to secrete extracellular vesicles that influence NK cells in a myriad of ways, including decreased levels of CD71 and CD98, impaired glucose uptake ability, and significant NKG2D, CD107a, TNF-α, and INF-γ downregulation. The ultimate result of these changes was a decreased NK cell cytotoxicity against cancer stem cells [30]. Ultimately, this specific immune cell population is the least well-studied with regard to the hepatic PMN, and thus, more research is necessary to uncover their further roles and whether it is synonymous with their establish role in cancer progression after dissemination. Unfortunately, to date, there has been a dearth of primary literature investigating the direct role of other lymphoid populations in the development of the hepatic PMN. Predominant emphasis has been placed on myeloid lineages, likely due to their upstream role in immunosuppression, as T cells are affected by the preceding actions of MDSCs. As the diverse web of immune cell interactions along with other actors in the PMN are unraveled, the authors suspect more understanding of B and T cell roles in the molecular mechanisms will be elucidated.

## 4. Liver Components

In addition to recruited BMDCs, native liver cell populations are the downstream effectors of primary tumor signaling. This is primarily mediated by stromal cells (hepatic stellate cells, Kupffer cells, and sinusoidal epithelial cells), but parenchymal hepatocytes have been shown to be involved as well. Moreover, the extracellular matrix (ECM) both serves as the physical bedrock for cellular activity and is also directly involved in pre-metastatic changes.

### 4.1. Stromal Components

#### 4.1.1. Hepatic Stellate Cells

Hepatic stellate cells (HSCs) are liver-specific pericytes located in the space of Disse that differentiate into highly proliferative and mobile myofibroblasts with diverse pro-inflammatory effects. While they are physiologically activated by liver injury and inflammatory signaling, tumor cells can also induce this change to promote metastatic colonization of the liver [69]. Upon pathologic activation, HSCs can promote tumor cell invasion by secreting MMPs to remodel existing ECM, producing additional collagen-rich ECM, and releasing ADAM9, which enhances tumor cell cleavage of surrounding laminin [70,71]. Furthermore, they have been implicated in angiogenesis, the recruitment of additional inflammatory cells, and immunosuppression of the tumor microenvironment by attenuating anti-tumor T-cell function [72,73,74]. 

The extent of their involvement in metastatic progression has motivated increasing attention to their influence at the pre-metastatic phase. Nielson et al. showed that murine PDAC tumors could recruit macrophages to the liver, which then influenced HSCs to produce a fibrotic microenvironment through periostin secretion [75]. Periostin is known to enhance metastatic growth in other primary tumor types [76,77], and granulin has been shown to exclude CD8+ T cells from liver metastases [78]; however, the study did not determine whether this mechanism is what led to successful metastatic colonization.

HSCs also promote the PMN by recruiting MDSCs to the liver. Under the influence of EVs derived from BMDCs in lung-cancer-bearing mice, they can remodel the ECM to be more susceptible to incoming MDSC migration [24]. In line with these findings, Grunwald et al. previously demonstrated that CXCR4+ neutrophil recruitment to the pre-metastatic livers of PDAC mouse models was mediated in-part by SDF-1 expression by activated HSCs [6]. Thus, HSCs mediate the dynamic recruitment of multiple immune cell types to the hepatic niche, where they can further prime the liver.

Recently, Zhao et al. demonstrated that HSCs could communicate back with the primary tumor following activation by miR-181a-5p released from CRC cells. Following EV uptake, HSCs facilitated CRC cell migration, invasion, and metastasis formation by increasing expression of α-SMA and fibronectin in the liver ECM while reducing vitronectin and tenascin C. Moreover, they also secreted CCL20 into circulation, which upregulated further miR-181a-5p production by primary tumor cells in a positive feedback loop [25]. This cross-communication between the PMN and the primary tumor expands past the notion of unidirectional signaling and offers additional targets for therapeutic blockade. In corroboration with the other studies mentioned, it also highlights the significant extent of HSC involvement in establishing the PMN necessary for future colonization and provides motivation for further study.

#### 4.1.2. Kupffer Cells

In contrast to the recruited macrophages described previously, Kupffer cells are the liver’s endogenous macrophage population that reside within the lumen of the sinusoids and act as the first line of defense against incoming insults. During metastasis, they serve a myriad of functions from circulating tumor cell elimination to promoting cell adhesion, invasion, and angiogenesis [79]. Ultimately, they have been shown to both promote and restrict metastatic growth depending on the stage of the colonization [80,81]. Interestingly, this contradiction has also been demonstrated with their involvement in the hepatic PMN. The work of Hoshino et al. and Shao et al. pointed to the pro-inflammatory capabilities of Kupffer cells. In both studies, exosomes released by primary malignancies stimulated Kupffer cell release of inflammatory mediators such as S100 proteins and IL-6 [26,27]. This could be due to S100-mediated upregulation of acute-phase response proteins serum amyloid A (SAA) 1 and SAA3. Hansen et al. demonstrated that these two proteins stimulate the release of multiple MMPs and cytokines, enhance breast cancer cell adhesion to fibronectin, and increase transcription of further S100 proteins to form a positive feedback loop [82]. Injection of mice with S100A4 and S100A8 increased the concentration of SAA1 and SAA3 in pre-metastatic livers, and it is possible S100 proteins upregulated by cancer do the same. *S100* gene expression has also been shown to correlate with clinical development of metastasis in patients [83]. In a separate study by Jiang et al., uptake of exosomes containing angiopoietin-like protein 1 (ANGPTL1) from CRC cells led to reduced Kupffer cell expression of MMP9, leading to reduced vascular leakiness within the liver. This was mediated by downregulation of the JAK2-STAT3 pathway and ultimately led to attenuation of downstream metastasis [28].

Ultimately, this incongruence underscores the complexity of Kupffer cell involvement in the PMN. It is likely that signaling from the primary tumor dictates the function they serve, but further study is necessary to elucidate this notion. Much like recruited macrophages, determining how to elicit the anti-metastatic phenotype of Kupffer cells could offer an intervention that promotes tumor cell clearance.

#### 4.1.3. Sinusoidal Endothelial Cells

Liver sinusoidal endothelial cells (LSECs) are a specialized population of discontinuous, fenestrated endothelial cells that form a barrier between the hepatic circulation and the underlying space of Disse and parenchyma. They are highly permeable, create low shear stress, and have a minimalistic basement membrane. Their unique nature allows them to play important roles in liver physiology, immunology, and pathology [84]. During metastasis, circulating tumor cells become entrapped within the narrow hepatic vasculature, wherein local Kupffer cells and NK cells can aid in cell clearance [85]. LSECs, however, express several surface proteins that can augment tumor cell adhesion, extravasation, and invasion into hepatic parenchyma. ICAM-1, E-selectin, and lectin have all been shown to interact with circulating tumor cells and mediate progression along the metastatic cascade [86,87,88].

In one study investigating LSEC pre-metastatic involvement, intestinal tumors led to significantly increased expression of apical fibronectin prior to tumor cell dissemination [89]. Additionally, Kim et al. showed that breast cancer could increase fibronectin expression in LSECs by secreting EVs containing TGF-β1 [29]. In both circumstances, upregulation of fibronectin increased cancer cell adhesion to liver sinusoids, thus promoting extravasation and advancing the metastatic cascade.

Moreover, VEGFA (known promoter of angiogenesis) was found to be upregulated in the pre-metastatic liver tissue of melanoma-bearing mice [90]. This correlated with altered microRNA expression patterns in the liver, suggesting a line of communication from the primary tumors that promotes vascular reorganization. Thus, LSECs have an established role in the hepatic PMN, and their further involvement in altered vascular permeability at this stage will be explored in a later section.

#### 4.1.4. Extracellular Matrix

The extracellular matrix (ECM) is a collection of proteins collectively referred to as the “matrisome” that serve several physiologic functions. The core matrisome comprises fibrillar proteins such as collagen, glycoproteins such as laminin, and the proteoglycans heparin sulfate and versican. Matrisome-associated proteins, on the other hand, are composed of proteins that process, remodel, and regulate the more traditional ECM components [91]. Collectively, the matrisome modulates the stiffness of the microenvironment, regulates mechanical signaling, expresses ligands for signal conduction, and directly sequesters and releases growth factors and chemokines [92]. Ultimately, the ECM represents a dynamic environment that is centrally involved in both physiologic and pathologic processes.

Significant evidence has implicated pre-metastatic tissue fibrosis as a promoter of metastatic colonization [93]. Within the liver of a breast cancer mouse model, Cox et al. demonstrated that lysyl oxidase (LOX) could establish a fibrotic microenvironment through crosslinking collagen and enhance metastatic colonization [94]. In a CRC model, peptidylarginine deiminase 4 (PAD4)-mediated citrullination of collagen I altered cancer cell adhesion and was essential for development of hepatic metastases [95]. One possible mechanism for this pro-metastatic effect of fibrosis was suggested by Erler et al. In this study, fibrosis increased the recruitment of myeloid cells that increased tumor cell invasion via MMP secretion [96]. 

Interestingly, the ECM changes induced by any given cancer line is specific to the organ of metastasis. Quantitative mass spectrometry analysis of metastases from the same breast cancer cell line to murine liver, lungs, brain, and bone marrow demonstrated distinct ECM changes depending on the context. In the hepatic niche, certain proteoglycans were uniquely upregulated compared to healthy tissue or other sites of metastasis [97]. While this may be intuitive considering the pre-existing variation in ECM composition, it corroborates the “seed and soil” hypothesis in which the influence and success of disseminated tumor cells depends on their destination. While a full discussion of ECM remodeling within the pre-metastatic window will be discussed in a later section, what is clear is that the dynamic and multifunctional nature of the ECM matrisome plays an integral role in the metastatic cascade.

### 4.2. Hepatocytes

Hepatocytes are the parenchymal cells of the liver and comprise the bulk of its mass. Despite their wide array of physiologic roles, their involvement in metastasis is relatively less understood compared to their stromal neighbors. In a murine model of breast cancer metastasis, they directly interacted with invading tumor cells via claudin-2 functioning as an adhesion molecule, and this was necessary for successful colonization [98]. Additionally, CRC cell incubation with hepatocyte-derived ECM proteins led to upregulation of tumor cell genes involved in migration, proliferation, communication, and angiogenesis [99]. These findings suggest that hepatocytes assist in remodeling the matrix towards a tumor-supportive composition, and this was later corroborated by Lee et al., who showed that hepatocytes facilitated fibrosis in the PMN after activation by PDAC-derived IL-6 [100]. Furthermore, lung-adenocarcinoma-derived EVs containing lncRNA-ALAHM could promote hepatic secretion of hepatic growth factor (HGF) [31]. HGF has been implicated in liver metastasis of several primary tumor types, and in this pre-metastatic setting, it promoted tumor cell proliferation, invasion, and migration. Thus, albeit less than other hepatic cell populations, hepatocytes have an established role in PMN priming.

## 5. Characteristics of the Hepatic PMN

The former sections reviewed signaling molecules and cell populations involved in formation of the hepatic PMN, but the crux is ultimately the specific changes within the liver that facilitate ensuing metastasis. The following discussion will review what is known about the liver PMN with regard to four principle PMN changes: ECM remodeling, inflammation, immunosuppression, and angiogenesis and increased vascular permeability. 

### 5.1. ECM Remodeling

Many of the paramount studies from the last decade that have begun to elucidate the mechanisms behind pre-metastatic priming have directly investigated ECM remodeling. A general theme has been the induction of liver fibrosis leading to immune cell recruitment, with the primary mediators being exosomes, as discussed in previous sections. In addition, they have been shown to cargo CCL2 and recruit macrophages to the liver, stimulating their differentiation into the immunosuppressive M2 phenotype and again increasing the downstream metastatic load [23]. Another demonstrated mechanism of fibrosis was mediated by exosomal CD44v6/C1QBP uptake by HSCs. In this PDAC model, exosomal uptake led to phosphorylation of insulin-like growth factor 1 signaling and ultimately HSC activation and fibrosis [32]. Beyond mediation by exosomes, Lee et al. demonstrated that IL-6 secreted from PDAC cells could activate STAT3 signaling in hepatocytes, which led to their deposition of collagen I and fibronectin. In combination with increased secretion of myeloid chemoattractant protein SAA, the result was an influx of MDSCs in addition to pancreatic liver metastases [100]. 

Collectively, these studies demonstrate a multitude of mechanisms by which primary tumors induce fibrosis within the pre-metastatic liver and link this change to immune cell recruitment and metastatic development. What remains to be explained is precisely how pre-metastatic fibrosis causes these downstream effects. The capabilities of specific immune populations were earlier reviewed, and their roles in primary tumor development and metastatic colonization have been studied, yet the methods by which they prime the pre-metastatic liver prior to dissemination are still unknown. Nonetheless, fibrosis is a known promoter of metastasis, and this is not exclusively contingent on primary tumor signaling. Patients with fibrotic livers according to the NAFLD fibrosis score (NFS) were shown to have increased hepatic recurrence of CRC following primary tumor resection [101]. Moreover, mice pre-treated with cisplatin were shown to develop liver fibrosis and increased hepatic metastasis following melanoma cell injection [102]. Thus, even considering its relatively large focus compared to other pre-metastatic liver hallmarks, ECM remodeling represents a highly complex area of tumor biology that is poorly understood. Its involvement in the studies described suggests a high level of impact, and multiple signaling molecules have already been identified for continued study. Because the downstream functions of recruited immune cells lead to the development of further pre-metastatic hallmarks such as inflammation and immunosuppression, these molecules could be key targets for therapeutic intervention.

### 5.2. Inflammation

It has long been known that inflammation is necessary for oncogenesis. Over a century ago, it was shown that chronic inflammation in mouse models could induce spontaneous tumors, and we now accredit this is to the milieu of cytokines, chemokines, and growth factors that support cancer development [103]. The mechanisms are analogous to the body’s response to tissue damage, so much so that tumors have even been described as “wounds that do not heal” [104]. A review of the specific mechanisms connecting tumorigenesis and metastasis were outlined by Hibino et. al, but a growing research effort has recently been focused on pro-inflammatory signaling prior to the onset of detectable metastases [105]. Regarding the liver, this pre-metastatic priming fosters an inflammatory environment that provides fodder for colonization of primary tumor cells upon extravasation into the parenchyma.

Multiple studies investigating the PMN have implicated interleukin 6 (IL-6), a pro-inflammatory cytokine known to have pathologic effects in the dysregulated processes of chronic inflammation and autoimmunity [106]. Shao et al. showed that Kupffer cells differentiated into a pro-inflammatory phenotype after uptake of CRC-derived EVs containing microRNA-21-5p. After differentiation, these Kupffer cells increased release of IL-6 and S100A8 in the pre-metastatic liver, which together led to increased systemic immune cell recruitment and resulting metastatic development [27]. In a separate murine model of CRC, EVs containing integrin beta-like 1 (ITGBL1) activated HSCs through TNFAIP3-mediated NF-kB signaling. The resulting upregulation of IL-6 and IL-8 correlated with increased frequency of fibronectin deposition, BMDC mobilization to the liver, and formation of CRC metastases in vivo [33].

More unorthodox methods have also been shown to be involved. In a mouse model of CRC, increased primary tumor disruption of the gut vascular barrier (GVB) allowed for bacterial translocation into circulation and eventually the liver. There, they increased transcription of several proinflammatory genes (*Saa 1/2/3/4*, *Tnfa*, *Ccl2*, and *Mmp15*) and mobilized distant macrophages and neutrophils to help establish a pre-metastatic inflammatory niche [107]. Additionally, liver ischemia/reperfusion injury (IRI) has been shown to promote neutrophil infiltration, neutrophil extracellular trap (NET) formation, endothelial adhesion molecule expression, and metastatic development [108,109,110]. Interestingly, an exercise training regimen prior to IRI was able to mitigate these pro-metastatic effects, suggesting an alternative form of non-pharmacologic therapy [111]. 

Ultimately, the mechanisms underlying inflammation are involved in many physiologic processes, and abrogating the pro-inflammatory support of tumor development is best stopped before it starts. Thus, pre-metastatic signaling presents an ideal window for intervention. Considering the Ji et al. study, targeted therapy against EV ITGBL1 might prevent their uptake and mitigate pre-metastatic inflammation altogether.

### 5.3. Immunosuppression

To successfully invade and colonize hepatic tissue, circulating tumor cells must overcome the persistent mechanisms of immunosurveillance that naturally defend against tumorigenesis [112]. By inducing immune tolerance within the liver prior to dissemination, primary cancers can thus increase their chances of survival once they spread. This process, known as immunosuppression, is carried out by several mechanisms. One of the most well-studied is the recruitment of MDSCs to the pre-metastatic liver. Connolly et al. demonstrated that intraperitoneal injection of CRC cells into mouse models induced the accumulation of MDSCs in the liver prior to the formation of micro-metastases. Mediated by primary tumor expression of keratinocyte-derive chemokine (KC), this increased metastatic colonization, and in vitro experiments suggested this could be due to inhibited T-cell proliferation and activation, prevention of CTL-mediated tumor lysis, and induction of Tregs [34]. Tregs have previously been shown to reduce cytotoxic T-cell efficacy [113] and enhance metastatic colonization [114], and it is likely they have a similar effect in the pre-metastatic context. 

A separate study showed that primary CRC tumors could induce MDSC liver accumulation via S1PR1-STAT3-IL6 signaling, ultimately leading to inhibited autologous T-cell proliferation in pre-metastatic models [35]. Furthermore, pre-metastatic myeloid cell accumulation can inhibit dendritic cell activation and induce macrophage differentiation towards the immunosuppressive M2 phenotype. This process was mediated by breast cancer secretion of GRP78 [36]. Secreted exosomes containing miR-135a-5p were even shown to directly inhibit CD30-mediated T-cell activation in murine CRC models, thus promoting immune tolerance and metastatic colonization of the liver [37].

Zhao et al. showed that exosomes likely impair NK cytotoxicity in the pre-metastatic liver as well. In vitro incubation of NK cells with PDAC-derived exosomes significantly downregulated their expression of NKG2D, CD107a, TNF-α, and INF-γ, leading to immune dysfunction. In vivo tracking of these exosomes upon injection in pre-metastatic mice demonstrated their uptake in the liver, suggesting they might localize to the hepatic microenvironment, where they then reduce NK-mediated tumor cell clearance [30]. Models of metastasis have further implicated HSCs and other mechanisms of M2 induction in hepatic immunosuppression; however, these mechanisms have yet to be demonstrated in pre-metastatic models [74,115].

One important consideration is that outside factors beyond primary tumors can prime the hepatic PMN. Bartlett et al. demonstrated that physiologic liver involution during weaning of post-partum mice was characterized by upregulation of the checkpoint molecule PD1 and an increased abundance of Tregs within hepatic tissue, ultimately conferring an increased susceptibility to breast cancer metastasis [116]. Although not due to primary cancer signaling, this change highlights the complexity of the PMN, which receives multiple inputs of influence and requires immense progress to better understand. With the fields of targeted therapy and immunotherapy continuing to burgeon, a better understanding of immunosuppression in the liver might uncover targets amenable to existing therapeutics. Thus, this hallmark of the PMN presents a particularly enticing option for clinical intervention.

### 5.4. Angiogenesis and Vascular Permeability

Just like healthy parenchyma, proliferating neoplasms depend on an adequate vascular network to deliver oxygen and other nutrients required for survival. Because they eventually expand beyond existing resources, neovascularization is a prerequisite for successful growth [117]. Moreover, vascular access is required for intravasation into the circulatory system, and the subsequent interaction with the endothelium of distant organs is equally important for extravasation [118]. It is now known that prior to metastatic spread, primary tumors can alter the vascular characteristics of other organs to augment their chances of seeding. While research on this concept regarding the liver is still in its infancy, recent studies have revealed evidence that pre-metastatic priming of hepatic sinusoids does occur. 

Zeng et al. demonstrated in vitro that exosomes containing microRNA-25-3p released from CRC cells could regulate the expression of VEGFR2, ZO-1, occludin, and claudin-5 in endothelial cells by targeting KLF2 and KLF4. The in vivo consequence of this effect was a dramatic increase in vascular permeability and angiogenesis in the liver along with an increase in metastatic colonization [38]. More recently, Yokota et al. found similar results using a hepatocellular carcinoma (HCC) model of intrahepatic spread. Here, HCC cells released several exosomal microRNAs that downregulated endothelial expression of ZO-1 and vascular–endothelial cadherin (VE-cadherin), resulting in increased vascular permeability and more intrahepatic metastases [39]. Far more knowledge on the processes of angiogenesis following metastatic seeding exists and has been extensively reviewed [119]; however, these two studies indicate that endothelial modification begins even earlier. Targeting this pre-metastatic priming could limit vascular permissiveness to incoming CTCs and inhibit later extravasation into the liver. If so, the metastatic cascade would be halted prior to more complex processes such as intrahepatic proliferation, immune evasion, dormancy, and reactivation, which could be more difficult to manage therapeutically.

## 6. Perspectives

While the field of the hepatic PMN is still in its early stages, the future of its clinical utility already looks promising. The following sections will cover emerging PMN targets for therapeutic intervention, plausible biomarkers to identify who would benefit, and new models to narrow the current gaps in knowledge. A summary of the relevant emerging therapeutics, biomarkers, and models for study can be found in Table 2.

### 6.1. Therapeutic Targets

At present, most investigations into PMN targeting have focused on the lung, but the strategies used offer opportunities for future study of the hepatic PMN. Ortiz et al. demonstrated that administration of reserpine to mice prevented melanoma-derived EV uptake in the lung and disrupted formation of the PMN [120]. While the same drug may not be applicable in liver metastasis, a similar pharmacologic screen might reveal options to reduce EV signaling from other primary tumors. In addition, exosomes have been used for targeted treatment delivery. In a study by Zhao et al., exosomes engineered to contain siS100A4 were injected into mice following primary breast cancer resection and shown to reduce the development of lung metastasis [121]. Moreover, exosomes have also been used to deliver chemotherapy to primary breast tumors with minimal toxicity in an integrin-specific manner [122]. Considering pancreatic exosomes demonstrated liver-specific uptake via αvβ5 expression, targeted therapy within the hepatic PMN context seems strongly feasible as well [26].

Other signaling pathways offer opportunities for intervention. In a pre-clinical model of pancreatic cancer, CXCR2 inhibition with pepducin depleted neutrophil and MDSC liver recruitment, suppressed metastases, and increased susceptibility to anti-PD1 therapy [10]. CXCR4-targeting has also been shown efficacious in multiple pre-clinical studies. Seubert et al. showed that CXCR4 inhibition with AMD3100 prevented neutrophil recruitment to the PMN, and Chen et al. found that AMD3100 injections decreased fibrosis and immunosuppression within the liver of breast-cancer-bearing mice [5,123]. As discussed previously, IL-6 has also been implicated in the hepatic PMN in several studies, and a phase I/II clinical trial has already shown the anti-IL-6 antibody siltuximab to be well-tolerated by patients with advanced solid tumors of varying origins [124]. The study did not demonstrate clinical activity of siltuximab, but that does not exclude its use in a separate context. Perhaps, earlier-stage patients would benefit through its use as an anti-PMN prophylactic that mitigates inflammation in the liver. At this stage, however, this use is still theoretical. 

Neutrophil extracellular traps (NETs) have also been under investigation for PMN targeting. Park et al. showed that nanoparticles coated with NET-digesting DNase I significantly reduced lung metastasis in mouse models of breast cancer [125]. Moreover, within the context of the liver, NETs not only trap CTCs within the sinusoids but also act as a chemotactic factor for cancer cells expressing the CCDC25 receptor. Knockout of CCDC25 abrogated breast/CRC metastasis to the liver, and serum NETs could even predict the occurrence of liver mets in early-stage breast cancer patients [126]. Thus, both targeting NETs for destruction via nanoparticles or inhibiting their signaling pathways offer viable opportunities for PMN prevention within the liver. 

A clinical trial currently under way is investigating the safety profile of dual administration of nivolumab (anti-PD-1 inhibitor) with galunisertib (TGFβ receptor I inhibitor) for advanced refractory tumors (NCT02423343). As previously discussed, TGFβ promotes inflammation, ECM remodeling, and immunosuppression within the hepatic PMN [5,29,55]. If galunisertib is shown to be well-tolerated, it might be used to prevent further metastatic development in stage IV patients or even to prevent initial metastases in earlier stages through PMN mitigation. Ultimately, however, like the suggested therapies above, this idea is still unproven.

### 6.2. Biomarkers

Despite the growing list of molecular players involved in the hepatic pre-metastatic niche, the field is young and has yet to produce a clinical biomarker amenable to assay for prognostic or predictive information. In theory, the secreted factors or circulating immune populations involved in PMN development could indicate looming metastatic progression, yet still even identifying the PMN with confidence in murine in vivo models is difficult given its spatiotemporal specificity. Exosomes containing unique microRNA signatures have shown promise as prognostic biomarkers for both gastric cancer [127] and CRC [128] patients across separate stages, and thus, they could circumstantially be thought of as PMN biomarkers, as levels would increase from early stage to metastasis, during which PMN development must occur. However, establishing biomarkers that directly indicate PMN formation will require significant more research.

Interestingly, an emerging field called “radiomics” has also offered new avenues of prognostic PMN biomarkers. This entails the use of medical imaging to mine quantitative data that provide clinical insight. In a recent study by Creasy et al., pre-operative CT imaging of stage II/III CRC patients before initial colon resection demonstrated significant differences in liver image qualities between patients who remained disease-free at 5 years and those who developed liver metastases [129]. Of note, decreased heterogeneity of hepatic tissue most strongly distinguished the recurrence group from those who had no evidence of further disease. These findings suggest that liver imaging even prior to initial primary tumor resection can identify patients at high risk of metastasis. This was corroborated by a multicenter study that found that radiomics analysis of primary staging CTs of CRC patients could predict liver metastasis with an AUC of 86% (95%CI 85–87%) [130]. Thus, imaging studies might one day be used to select high-risk patients for early intervention. Most likely is that a combination of clinical factors, liquid biopsy data, and imaging will be used in coordination.

### 6.3. Emerging Methods and Models

The hepatic PMN is a particularly difficult phenomenon to study due to its temporospatial specificity. It marks a period of tissue change prior to the arrival of disseminated tumor cells, but demonstrating this experimentally through in vivo models is difficult. Present studies typically start by identifying primary tumor signaling molecules of interest, conditioning mice with these molecules, investigating pre-metastatic liver changes, and then challenging with tumor cell injection to determine increased susceptibility to metastasis [38]. While such techniques have led to the current body of knowledge, the emergence of new methods of study will hopefully better elucidate the chronicity and mechanisms involved.

Three-dimensional tissue models present an area of promise. Carpenter et al. implanted hydrogel scaffolds seeded with human stromal cells into prostate cancer mouse models to serve as proxy PMNs. By imaging microenvironments within the scaffolds, they were able to monitor interactions among disseminated tumor cells, leukocytes, stromal cells, and endothelial cells [131]. Moreover, by monitoring changes in these interactions as tumor cell populations grew, the model gave insight into the cellular interactions governing micro-metastatic progression. 

Additionally, scaffolds can be prepared from native patient samples through decellularization, which removes the cellular components from tissue and leaves behind the ECM. In a study by Persson et al., ECM scaffolds made from patient breast cancer samples were recellularized with cell lines to assess how established tumor-microenvironments affect cancer progression [132]. When considering this methodology with that of Carpenter et al., in theory, a human liver scaffold could be implanted into mice bearing CRC tumors and monitored for cellular interactions within the PMN. Decellularization has also allowed for protein characterization of ECM samples and could be used to better assess early liver ECM changes during the metastatic cascade of mouse models [134].

Spheroid models of CRC have also been used to study the in vitro upregulation of certain signaling pathways during tumor proliferation [133]. This might shed light on new signaling targets involved in PMN formation. Furthermore, 3D microfluidic “liver-on-a-chip” models provide an ideal system to study communication between separate organ components. Kim et al. utilized this method to show that breast cancer-derived EVs stimulated LSECs to upregulate fibronectin and break down cell barriers [29]. Ultimately, using different monitoring techniques, this microphysiological system could offer insight into primary tumor signaling mechanisms along with the chronicity of change within the liver compartment relative to tumor cell invasion. This makes it well-suited for future investigations on the PMN.

## 7. Conclusions

Although Paget originally proposed his “seed and soil” theory in 1889, research supporting the pre-metastatic niche has only truly materialized within the last two decades. In large part, this body of work has focused on the lung PMN, but the hepatic PMN has garnered increasing interest as of late. We now know that dynamic interactions between primary tumor signaling, recruited BMDCs, and native liver characteristics lead to a set of defined changes that foster metastatic growth. Inhibiting the metastatic cascade at this point, prior to the onset of colonization, presents an attractive yet elusive opportunity. Ultimately, this would require both a PMN-specific target and a well-tolerated therapeutic that could be administered long-term in early-stage patients prior to the onset of metastasis. Nonetheless, the need for such treatment is large. Metastasis is the true killer of cancer, and we currently have few strategies to stop its development rather than treat its existence. Hopefully the emergence of the PMN will offer an opportunity to change this paradigm and alleviate the burden of patients who often need it the most.

## Figures and Tables

**Figure 1 cancers-14-03731-f001:**
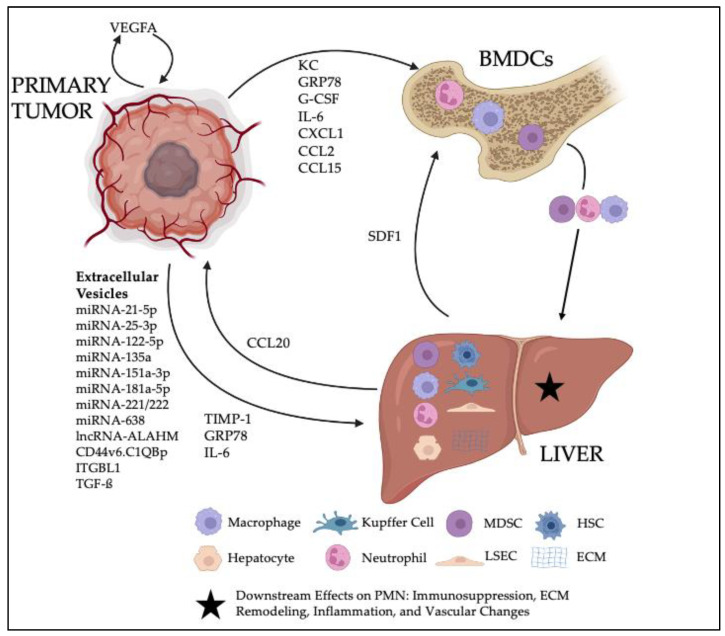
Scheme demonstrating the tissues, cells, messengers, and changes involved in orchestrating the hepatic PMN. The primary tumor secretes molecular communicators into circulation, which interact with bone marrow immune populations as well as directly with resident cells in the liver. The resulting interactions among all these components is a set of characteristic changes that define the development of the PMN and facilitate susceptibility to downstream colonization.

**Table 2 cancers-14-03731-t002:** Summary of future perspectives regarding the hepatic PMN.

Category	Technique	Primary Tumor	Metastatic Site	Mechanism	References
Therapeutic	Reserpine	Melanoma	Lung	Reserpine prevented melanoma-derived EV uptake in the lung	[120]
Therapeutic	Exosomal si100A4	Breast	Lung	Exosomes loaded with si100A4 reduced the development of metastasis following tumor resection	[121]
Therapeutic	Exosomal doxorubicin	Breast	n/a	Exosomes delivered doxorubicin to breast tumors in an integrin-specific manner, leading to minimal toxicity	[122]
Therapeutic	Pepducin	Pancreatic	Liver	CXCR2 inhibition depleted neutrophil and MDSC liver recruitment, leading to decreased metastasis	[10]
Therapeutic	AMD3100	CRC	Liver	AMD3100 injections prevented neutrophil recruitment to liver PMN	[5]
Therapeutic	AMD3100	Breast	Liver	AMD3100 decreased fibrosis and immunosuppression in liver	[123]
Therapeutic	Siltuximab	Various	n/a	Phase I/II clinical trial demonstrating siltuximab (anti-IL-6 antibody) to be well-tolerated	[124]
Therapeutic	Nanoparticle DNase I	Breast	Lung	DNase I injected via nanoparticles significantly reduced metastasis	[125]
Therapeutic	Anti-CCDC25 antibody	Breast, CRC	Liver	Anti-CCDC25 antibody targeting NETs led to decreased breast and CRC metastasis	[126]
Therapeutic	Nivolumab + Galunisertib	Various	n/a	Clinical trial investigating safety profile of galunisertib (TGFß receptor I inhibitor) administered with nivolumab (PD-1 inhibitor)	NCT02423343
Biomarker	microRNA-23b	Gastric	n/a	Prognostic biomarker associated with individual stages of gastric cancer progression	[127]
Biomarker	microRNA-548-5p	CRC	n/a	Prognostic biomarker associated with individual stages of CRC progression	[128]
Biomarker	Pre-operative CT	CRC	n/a	Radiomic analysis of CT imaging prior to primary tumor resection predicted recurrence	[129,130]
Model	3D tissue model	Prostate	n/a	Hydrogel scaffolds seeded with stromal cells to serve as a proxy for the PMN	[131]
Model	Decellurized scaffold	Breast	n/a	Decellurized ECM scaffolds from breast tumor samples then recellularized with cell lines to assess tumor microenvironment	[132]
Model	Spheroid	CRC	n/a	Spheroid models used for in vitro analysis of signaling pathways	[133]

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
