# Peer review of "The Hepatic Pre-Metastatic Niche"

_cancers, 2022, doi:10.3390/cancers14153731_

Round 1

Reviewer 1 Report

The following manuscript “The Hepatic Pre-Metastatic Niche” by Ormseth et al summarizes current knowledge about the liver as a pre-metastatic niche and its mechanisms. The manuscript is generally well written and offers comprehensive up to date information about the topic. However, a few issues need to be revised before recommending publication.

Major things:

The structure of the review is a bit confusing. Majority of chapter 4 “Characteristics of the Hepatic PMN” overlaps with the former chapters, as the topics are difficult to separate. I would suggest moving complete text from chapter 4 to respective former chapters (for example parts about exosomes better fit to 1.2 Exosomes, parts about fibrosis to 3.1.4 Extracellular matrix, etc.). This will greatly reduce redundancy and make it easier to follow the red line.

The authors strongly focused on the influence of immune cells from the myeloid lineage on the hepatic PMN with low information about cells from the lymphoid lineage, which are also very important. While I think this is no problem, they should mention that fact in their manuscript and reference current literature also reviewing the influence of the lymphoid lineage.

The figure is not totally self-explanatory. What is the difference between the factors on the left side and the factors in the box on the right side of the figure? Especially having Table 1, which gives a great overview, I feel most of the molecule, cytokine etc. names in figure 1 should be deleted. Instead, mechanisms of action (like vascular remodeling, immunosuppression etc.) should be annotated to the elements or arrows in the figure.

Reviewer 2 Report

I.    Summary

The review article “The Hepatic Pre-Metastatic Niche” aims to summarize the current state of knowledge of the concept of the pre-metastatic liver, the favorable molecular landscape which encourages hepatic metastazation of various primary cancers. This review covers the key signaling molecules, cellular components and characteristic changes including ECM remodeling, inflammation, immunosuppression and angiogenesis, involved in the emergence of the hepatic PMN microenvironment. Ultimately, the authors attempt to point out the eminent relevance of further research and emerging of useful preclinical models for hepatic PMN to determine PMN targets and biomarkers which will become significant for clinical utility and new treatment modalities.

II.   General concept comments

With providing a synthesis about the concept of the pre-metastatic liver, the review article addresses a highly relevant topic in cancer biology and cancer patient treatment. Overall, the manuscript is written in excellent English, very well structured and comprehensive. Figure and Table are appropriate and easy to understand and require only small changes (see specific comments below).

Worthy of note, there have been published several review articles addressing the hepatic pre-metastatic niche since 2016. The authors basically present a concised version of the review from Azizidoost et al. (doi: 10.1007/s13277-015-4557-x). Therefore, the novelty of this article is disputable.

1.    What distinguishes the hepatic PNM from other PMNs? (patient survival etc)

2.    Limitations/gaps of knowledge?

3.    Are there any risk factors that promote the formation of hepatic PMN?

4.    Please add a paragraph summarizing primary tumors/cancer types that favor explicitly the hepatic PMN for metastatic seeding

Epidemiology of hepatic PMN

III. Specific comments

1.    Page 1, line 29: Please give the full term “Gastrointestinal“ before “GI”.

2.    Page 2, figure 1: For clarity reasons and to highlight the interaction of three entities orchestrating the PMN, please add the terms “Primary tumor”, “BMDCs” and “Liver” next to the corresponding schematic illustrations.

3.    Page 2, Figure 1:

·         Does “EVs” stand for “extra-cellular vesicle”? Please clarify the abbreviation in the caption.

·        4.    Page 7, line 213: Please add the term “tumor-associated macrophages” before “TAMs” for clarification.

Reviewer 3 Report

Manuscript by Ormseth et al comprehensively summarized the field of liver pre-metastatic niche. Overall, the manuscript is well-written and ideas are logically flow. Minor revision is recommended prior acceptance for publication:

1) Please define what is CRC before abbreviating (Page 3 Line 84)

2) Change motivate to motivation (Page 3 Line 82) (Page 8 Line 310)

3) Expand a little more on the role of G-CSF in breast cancer (Page 3 Line 93)

4) There are some literatures on the role of NK cells in pre-metastatic niche. Would be great to add this information to Section 2 of the review.

5) Please name the type of specific proteoglycans (Page 10 Line 379)

6) Would be great to have a summarizing figure for the perspective section to help readers to have a quick overview.

Round 2

Reviewer 1 Report

Answer to response 1:

I agree with the authors, that the mentioned review “Characteristics and Significance of the Pre-metastatic Niche” (2016) by Yang Liu and Xuetao Cao did a great job in using a similar outline but keeping the level of redundancy to a minimum. However, I disagree, that the authors of this manuscript also succeeded with that approach. The clear separation of the different paragraphs is not obvious and there is a high level of redundancy. One of the most obvious examples is the description of exosome release/Kupffer cell impairment/ECM remodeling/increased metastasis. That whole paragraph is re-written in 1.2. Exosomes , 3.1.2. Kupffer cells , 4.1. ECM remodeling and 4.2 Inflammation. Thus, if the authors insist on keeping the structure, they need to clearly separate each point from the other.

My other major points were satisfactorily addressed.
